# Piperacillin/tazobactam resistance in a clinical isolate of *Escherichia coli* due to IS26-mediated amplification of *bla*$_{TEM-1B}$

Alasdair T. M. Hubbard [1]✉, Jenifer Mason[2], Paul Roberts [2,7], Christopher M. Parry[3,4,5,6], Caroline Corless[2], Jon van Aartsen [2], Alex Howard[2], Issra Bulgasim[1], Alice J. Fraser[1], Emily R. Adams[1], Adam P. Roberts [1] & Thomas Edwards [1]✉

A phenotype of *Escherichia coli* and *Klebsiella pneumoniae*, resistant to piperacillin/tazobactam (TZP) but susceptible to carbapenems and 3rd generation cephalosporins, has emerged. The resistance mechanism associated with this phenotype has been identified as hyperproduction of the β-lactamase TEM. However, the mechanism of hyperproduction due to gene amplification is not well understood. Here, we report a mechanism of gene amplification due to a translocatable unit (TU) excising from an IS26-flanked pseudo-compound transposon, PTn6762, which harbours *bla*$_{TEM-1B}$. The TU re-inserts into the chromosome adjacent to IS26 and forms a tandem array of TUs, which increases the copy number of *bla*$_{TEM-1B}$, leading to TEM-1B hyperproduction and TZP resistance. Despite a significant increase in *bla*$_{TEM-1B}$ copy number, the TZP-resistant isolate does not incur a fitness cost compared to the TZP-susceptible ancestor. This mechanism of amplification of *bla*$_{TEM-1B}$ is an important consideration when using genomic data to predict susceptibility to TZP.

[1] Department of Tropical Disease Biology, Liverpool School of Tropical Medicine, Pembroke Place, Liverpool L3 5QA, UK. [2] Liverpool University Hospital Foundation Trust, Prescot Street, Liverpool L7 8XP, UK. [3] Alder Hey Children's NHS Foundation Trust, Eaton Road, Liverpool L12 2AP, UK. [4] Department of Clinical Infection, Microbiology and Immunology, University of Liverpool, Liverpool L69 7BE, UK. [5] Clinical Sciences, Liverpool School of Tropical Medicine, Pembroke Place, Liverpool L3 5QA, UK. [6] School of Tropical Medicine and Global Health, University of Nagasaki, Nagasaki, Japan. [7] Present address: Faculty of Science and Engineering, University of Wolverhampton, Wulfruna Building MA, Wulfruna Street, Wolverhampton WV1 1LY, UK. ✉email: alasdair.hubbard@lstmed.ac.uk; thomas.edwards@lstmed.ac.uk

β-lactam/β-lactamase inhibitor combinations were developed to overcome the activity of β-lactamases[1,2], which inactivate β-lactam antibiotics by hydrolysing the β-lactam ring. β-lactamase inhibitors, particularly metallo-β-lactamase inhibitors, are still urgently required and their discovery and development is the topic of intense investigation[3]. β-lactam/β-lactamase inhibitor combinations currently in clinical use include amoxicillin/clavulanic acid, ampicillin/sulbactam and piperacillin/tazobactam (TZP), with ceftolozane/tazobactam and ceftazidime/avibactam recently introduced into clinical use. TZP has broad-spectrum antibacterial activity and is routinely used for intra-abdominal infections and febrile neutropenia[4,5]. TZP usage has increased year on year in the UK, from just under 2.1% of all antibiotics prescribed in 2008–2009 to 3.6% in 2012–2013[6]. During the 2-year period between April 2012 and March 2014 10.2% of bacteraemia causing *Escherichia coli* isolates in England tested for TZP susceptibility were resistant[7]. Resistance to TZP has been previously linked to AmpC hyperproduction and the co-production of multiple β-lactamases, which also confer resistance to 3rd generation cephalosporins[8]. In addition, tazobactam is a poor inhibitor of metallo-β-lactamase enzymes, which cause resistance to TZP, alongside 3rd generation cephalosporins and carbapenems[9–11].

A phenotype in *Klebsiella pneumoniae* and *E. coli* clinical isolates has emerged which has been classified as TZP-non-susceptible but susceptible to 3rd generation cephalosporins and carbapenems[12–14], indicating a different resistance mechanism. While still relatively rare, one study in the United States found that the frequency of this phenotype was between 1.9 and 5.6% of *E. coli* and *K. pneumoniae* isolated from the bloodstream between 2011 and 2015, and specifically 4.1% of all *E. coli* over the study period[14]. The same study reported that risk factors associated with the TZP-non-susceptible but 3rd generation cephalosporin and carbapenem susceptible phenotype included exposure to β-lactam/β-lactamase inhibitors and cephalosporins within the previous 30 days[14]. Resistance to TZP, but 3rd generation cephalosporin and carbapenem susceptible, has been linked to the presence of β-lactamases which hydrolyse piperacillin but not 3rd generation cephalosporins. The β-lactamases such as SHV-1 and TEM-1 are usually inhibited by tazobactam, which also has an intermediate inhibitory activity towards OXA-1[15,16]. TEM has been hypothesised to overcome the inhibitory activity of tazobactam via hyperproduction of the enzyme, allowing the hydrolysis of piperacillin[17]. Mechanisms leading to hyperproduction include mutations in the promoter region of $bla_{TEM}$, changing it from a weak promoter (*P3*) to a stronger promoter (*P4* or *P5*)[18] or a single point mutation further upstream resulting in the overlapping, stronger promoter *Pa/Pb* superseding the weaker *P3* promoter[19], increasing the production of TEM. Another such mechanism proposed to cause TZP-resistance but 3rd generation cephalosporin and carbapenem susceptibility is the increase in copy number of $bla_{TEM}$ present in either a plasmid or chromosome[17,20]. Gene amplification has been linked to the cause of temporary antibiotic resistance seen in a sub-population of bacteria and is known as heteroresistance. Heteroresistance is often lost after multiple generations in the absence of antibiotic selective pressure, due to the fitness cost imposed by the production of extra proteins as a result of amplification[21,22]. While the mechanism of amplification of $bla_{TEM}$ is not well known, recent studies have found that the amplified $bla_{TEM}$ has been co-located on a segment of DNA containing other antibiotic resistance genes, such as *aadA* and *sulI*, termed a genomic resistance module[17]. Amplification of $bla_{TEM}$ leading to TZP resistance via β-lactamase hyperproduction has also been suggested to be mediated by the presence of the insertion sequence, IS*26*[20]. IS*26* is often linked with the movement of antibiotic resistance genes; for example a translocatable unit (TU) containing IS*26* has been shown to be able to excise from the transposon Tn*4352*B, which itself was located on a plasmid,

between two IS*26*, leaving one in the plasmid[23,24]. Recently, directly repeated IS*26* elements, plus the intervening DNA, have been labelled as pseudo-compound transposons (PTns)[25] rather than composite transposons as they do not transpose as an entire unit. Following excision, the single IS*26* and antibiotic resistance gene(s) found between the two insertion sequences forms a circular TU, which then can insert into a plasmid via a conservative Tnp*26*-dependent but RecA-independent mechanism, Tnp*26* replicative transposition or RecA-dependent homologous recombination, preferentially adjacent to another IS*26* insertion sequence[23,24,26].

Here, we identify a pair of clonal *E. coli* isolates, isolated from a single patient across two separate infection episodes, which display within-patient evolution to TZP resistance. In this isolate, amplification of $bla_{TEM-1B}$ occurs when a TU excises from a pseudo-compound transposon flanked by directly repeated IS*26* present in the chromosome. The TU re-inserts into the chromosome creating a tandem array of the TU and increasing the copy number of $bla_{TEM-1B}$, which does not carry a fitness cost. Replicating the evolutionary event in vitro in the TZP-susceptible isolate leads to the capture of the TU in a plasmid which contains a copy of IS*26*.

## Results

**Identification of clonal isolates.** Initially, we identified five isolates in the collection of TZP-resistant, 3rd generation cephalosporin and carbapenem susceptible *E. coli* from blood cultures at the Royal Liverpool University Hospital (RLUH) which had a corresponding TZP-susceptible isolate from the same or previous infection episode, and therefore could have evolved to become TZP-resistant within a patient. Restriction fragment length polymorphisms (RFLP) of the 16S rRNA amplicons from the five pairs of isolates indicated that three pairs of TZP-susceptible/TZP-resistant clinical isolates had identical digestion patterns (Supplementary Fig. 1A). Two of these three pairs of isolates had an identical resistance profile generated during routine disk-based susceptibility testing, aside from TZP (Supplementary Table 1). RFLPs of genomic DNA identified one pair of isolates with identical banding patterns indicating clonality; 190693 (TZP-susceptible) and 169757 (TZP-resistant) which were isolated from different infection episodes from the same patient ~3 months apart (Supplementary Fig. 1B). During the first infection episode, the TZP-susceptible *E. coli* was isolated and the patient was initially treated with a five-day course of TZP, followed by a seven-day course of TZP with teicoplanin and then a third seven-day course of TZP although a second blood culture was found to be negative. A second infection episode occurred ~6–7 weeks after the final course of TZP was completed, and again the patient was treated initially with TZP until the TZP-resistant *E. coli* was isolated, when the treatment was changed to meropenem. Putative clonality of these two isolates was confirmed with whole-genome sequencing; both isolates were identified as serotype H30 O86, sequence type 315 and had an average nucleotide identify (ANI) of 100%, with 36 single nucleotide polymorphisms (SNP) difference between the two isolates.

**Confirmation of TZP susceptibility and resistance mechanism.** We determined the minimum inhibitory concentrations (MIC) of the pair of isolates and verified that TZP-susceptible isolate was susceptible to TZP (2–4/4 μg/ml) and TZP-resistant isolate was resistant to TZP (64/4 μg/ml) according to European Committee on Antimicrobial Susceptibility Testing (EUCAST) clinical breakpoints[27] (Table 1). Using the efflux pump inhibitor phenylalanine-arginine β-naphthylamide (PAβN) as a supplement in the MIC assay, we were able to rule out overexpression of efflux pumps as a possible mechanism of resistance as there was less than a fourfold reduction[17] in MIC of both the TZP-susceptible (2/4 μg/ml) and TZP-resistant isolates (32/4 μg/ml, Table 1). Whole-genome

**Table 1 Minimum inhibitory concentrations of the piperacillin/tazobactam-resistant and piperacillin/tazobactam-susceptible isolates.**

|  | GEN | TET | CHL | CIP | AMC | TZP | TZP + PAβN |
|---|---|---|---|---|---|---|---|
| TZP-susceptible | 128 μg/ml | 256 μg/ml | 4 μg/ml | 64 μg/ml | 32 μg/ml | 2–4/4 μg/ml | 2/4 μg/ml |
| TZP-resistant | 1024->1024 μg/ml | 512 μg/ml | 4 μg/ml | 128 μg/ml | 64–128 μg/ml | 64/4 μg/ml | 32/4 μg/ml |

Minimum inhibitory concentrations (MIC) of gentamicin (GEN), tetracycline (TET), chloramphenicol (CHL), ciprofloxacin (CIP), amoxicillin/clavulanic acid (AMC) and piperacillin/tazobactam (TZP) (with and without Phenylalanine-arginine β-naphthylamide (PAβN)) towards the TZP-susceptible and TZP-resistant isolates. Determination of MICs were performed in triplicate ($n = 3$).

**Table 2 Predicted antimicrobial resistance genes.**

| Antimicrobial resistance gene | TZP-susceptible isolate | TZP-resistant isolate |
|---|---|---|
| $bla_{OXA-1}$ | Chromosome | Translocatable unit |
| $bla_{TEM-1B}$ | Chromosome | Translocatable unit |
| aac(3)-lla | Chromosome | Translocatable unit |
| aac(6')-lb-cr | Chromosome | Translocatable unit |
| aadA1 | Chromosome | Chromosome |
| aph(3")-lb | Chromosome | Chromosome |
| aph(6)-ld | Chromosome | Chromosome |
| tet(D) | Chromosome | Translocatable unit |
| dfrA1 | Chromosome | Chromosome |
| sul2 | Chromosome | Chromosome |
| mdf(A) | Chromosome | Chromosome |
| catB3 | Chromosome | Translocatable unit |

Predicted antimicrobial resistance genes by ResFinder found on the genome of the piperacillin/tazobactam (TZP)-susceptible and TZP-resistant isolates and their position in genome. The resistance gene catB3 was predicted by ResFinder to be present with 69.8% length, but both isolates were phenotypically chloramphenicol susceptible.

sequencing revealed no differences in the predicted resistance genes present in the genome between the TZP-susceptible and TZP-resistant isolate (Table 2) and no mutations in the promoter region of any of the β-lactamases present within the genome ($bla_{TEM-1B}$, $bla_{OXA-1}$ or $ampC$) of the TZP-resistant isolate. We confirmed that the TZP-resistant isolate hyperproduced a β-lactamase due to the significant increase in nitrocefin hydrolysis compared to the TZP-susceptible isolate ($P$ value = <0.0001, Fig. 1a, Supplementary Table 3). The isolates contained two β-lactamases, TEM-1B and OXA-1, which were both increased in copy number in the TZP-resistant isolate compared to the housekeeping gene $uidA$ (Fig. 1b). We, therefore determined whether TEM-1B, OXA-1 or both contributed to TZP resistance using increased amount of tazobactam to inhibit TEM-1B and 100 mM sodium chloride to inhibit OXA-1[28–30]. Increasing tazobactam from 4 to 8 μg/ml and 4 to 16 μg/ml resulted in a twofold and eightfold decrease in TZP MIC respectively, while supplementation of 100 mM sodium chloride decreased the TZP MIC by twofold (Supplementary Table 2). As there was only a modest decrease in resistance in the presence of sodium chloride, it is likely hyperproduction of TEM-1B is the most important determinant of resistance to TZP.

**Genomic comparison of the clonal isolates**. Using a hybrid assembly of Oxford Nanopore Technologies (ONT) long and Illumina short sequencing reads, we were able to complete the genome of the TZP-susceptible isolate, which was found to be 5151952 bp in length, with a GC content of 50.64% and did not contain any plasmids (Supplementary Fig. 2A). In contrast, we were unable to complete the genome of the TZP-resistant isolate as a 530 bp segment remained unresolved, and a complete, low copy number (2.59x) 106637 bp plasmid containing an IncFII replicon (Supplementary Fig. 2B) was detected. A complete, smaller (10899 bp) circular DNA molecule was also found to be present in the

TZP-resistant isolate, at a copy number of 8.51x, however, this small circular DNA molecule did not contain a plasmid replicon (Supplementary Fig. 2B). The large plasmid did not to contain any predicted antimicrobial or metal resistance genes, but did contain three bacteriocins, both colicin B and M (with cognate immunity proteins) and linocin. Comparison of the predicted resistance genes present on the chromosome of the TZP-susceptible and TZP-resistant isolates highlighted that $bla_{TEM-1B}$, $bla_{OXA-1}$, aac(3)-lla, aac(6')-lb-cr, tet(D) and catB3 were missing from the assembled chromosome of the TZP-resistant isolate (Table 2). Characterisation of the small circular DNA molecule found that it contained these missing resistance genes, as well as several putative transposable elements including three copies of IS26 (Table 2), and aligned exactly to the chromosome of the TZP-susceptible isolate and was no longer present in the chromosome of the TZP-resistant isolate. The predicted catB3 resistant gene was truncated to 69.8% length and therefore unlikely to be functional, which was confirmed as both the TZP-susceptible and TZP-resistant isolates were susceptible to chloramphenicol (CHL) according to EUCAST clinical breakpoints (Table 1). Further analysis of the TZP-susceptible genome uncovered that the circular DNA molecule from the TZP-resistant isolate aligned with 100% identity to an integrated pseudo-compound transposon flanked by two copies of IS26 in the same orientation, which we subsequently registered as PTn6762 via The Transposon Registry[31], however, the circular DNA molecule only contained one of the flanking IS26. The antibiotic resistance genes $bla_{TEM-1B}$, $bla_{OXA-1}$, aac(3)-lla, aac(6')-lb-cr and tet(D) were identified to be present on PTn6762 (Fig. 2a). This suggested that a TU[23,24,26], containing one flanking IS26 and the antibiotic resistance genes, was excised from the chromosomally located PTn6762 while the other flanking IS26 stayed in the chromosome (Fig. 2a). Interestingly, we also found that the tetracycline resistance regulator, tetR, on the TU had been disrupted 52 bp from the end of the tetR due to the excision which overlaps the start of the copy of IS26 remaining in the chromosome. Following excision, a copy of IS26 is present at the start of the TU which then connects to the end of the TU, containing the disrupted tetR, forming a circular DNA molecule. As the two copies of IS26 are identical and in the same orientation, and therefore containing the same 52 bp bases missing from tetR, tetR was reformed when the TU circularised completing the 657 bp gene.

**Confirmation of the amplification of the TU**. Firstly, we confirmed the amplification of the resistance genes found on the TU in the TZP-resistant isolate and PTn6762 in the chromosome of the TZP-susceptible isolate. Comparing the fold change in copy number of each resistant gene to the housekeeping gene $uidA$, we found that each resistance gene on the TU increased in copy number in the TZP-resistant isolate compared to the TZP-susceptible isolate ($P$ value = <0.0001, Fig. 1b, Supplementary Table 3). The increase in copy number of the resistance genes found on the TU in the TZP-resistant isolate also corresponded to an increase in MIC of all antimicrobials that the genes confer resistance to (Table 1), further confirming the amplification of the

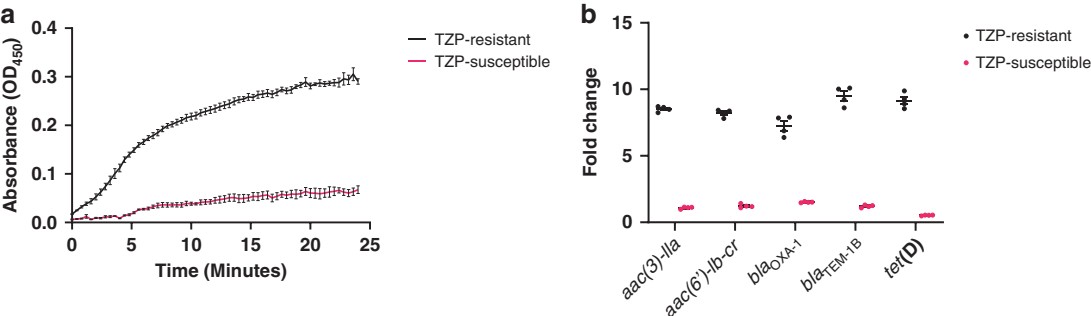

**Fig. 1 Hyperproduction of TEM-1B in the piperacillin/tazobactam-resistant isolate is mediated by an increase in copy number. a** Increase in nitrocefin hydrolysis by the piperacillin/tazobactam (TZP)-resistant isolate (black line) compared to the TZP-susceptible (pink line) isolate, as measured at an optical density of 450 nm ($OD_{450}$), due to hyperproduction of a β-lactamase in the TZP-resistant isolate. All error bars represent the standard error of the mean ($n = 3$). **b** Comparison of the fold change in copy number of the antimicrobial resistance genes present on the pseudo-compound transposon of the TZP-susceptible isolate (pink circles)/translocatable unit (TU) of the TZP-resistant isolate (black circles) as assessed by qPCR of a single DNA extract compared to the housekeeping gene *uidA*. All error bars represent the standard error of the mean ($n = 4$). Source data are provided as a Source Data file.

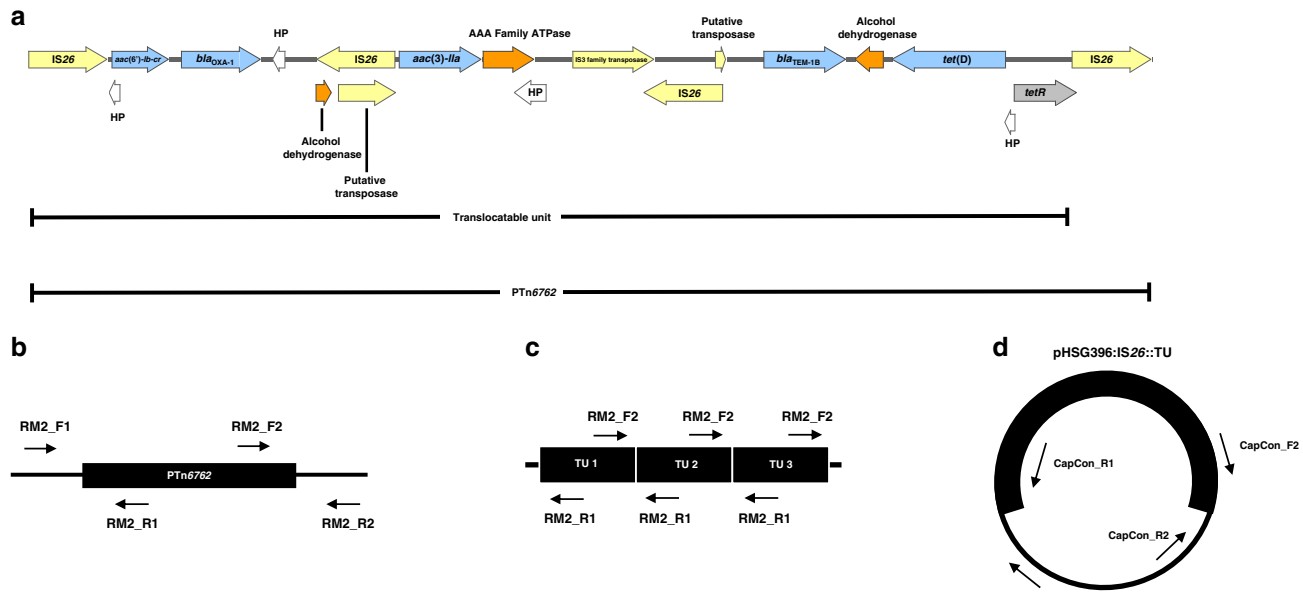

**Fig. 2 Schematics of the pseudo-compound transposon PTn*6762*, translocatable unit and positions of primers used in this study.** Schematic showing (**a**) the characterisation of PTn*6762* (HP = hypothetical protein) and the position of the primer pairs to detect (**b**) the junctions of the PTn*6762* in the chromosome, (**c**) the presence of the tandem array of TUs in the chromosome and (**d**) the junctions of the insertion of the TU into pHSG396:IS*26*.

entire TU. By PCR of the left and right junctions of the chromosomally located PTn*6762*, with one primer specific for PTn*6762* and one for the chromosome either before or after IS*26* (Fig. 2b, Supplementary Table 4), we were able to confirm that PTn*6762* was present in the chromosome of both the TZP-susceptible isolate and TZP-resistant isolate by yielding the expected 1640 bp and 2402 bp products, respectively (Supplementary Fig. 3). Using primers that would only yield a 1942 bp product if a tandem TU array in the chromosome was present (Fig. 2c), we were able to detect the presence of the tandem TU in the TZP-resistant isolate and absence in the TZP-susceptible isolate (Supplementary Fig. 3). To further confirm that the TU is present in the chromosome in tandem arrays, we mapped the long-read sequences to a predicted structure of three tandem TUs and built a consensus sequence out of these reads. We were unable to build a consensus sequence for the TZP-susceptible isolate, suggesting the TU was not present in the chromosome in a tandem array. In contrast, we were able to build a consensus sequence for the TZP-resistant isolate, which showed 99.89% identity to the predicted sequence (Supplementary Fig. 4). This suggests that the TU from PTn*6762* has excised and reinserted to form a tandem array within the chromosome of the TZP-resistant isolate.

**Capture of the TU**. We sought to capture the excised TU, and therefore observe the excision and insertion events, using a CHL resistant, high copy number pUC plasmid (pHSG396) containing IS*26* in the same orientation relative to the origin of replication as found on PTn*6762* (Supplementary Methods 1). pHSG396:IS*26* was transformed into the TZP-susceptible isolate and TZP-resistant derivatives were selected for by growing the isolate in the presence of TZP. We detected the insertion of a >10 kb fragment into pHSG396:IS*26* after TZP selection following digestion with XhoI and EcoRI (Supplementary Fig. 5A). Insertion of the TU from the TZP-susceptible chromosome was confirmed through PCR amplification across the two newly formed junctions on the pHSG396:IS*26*, with one primer specific for pHSG396 and one for either *aac(6')-Ib-cr* (left) or *tet*(D) (right) on the TU for each junction (Supplementary Table 4). We yielded the expected 1458 bp (left)

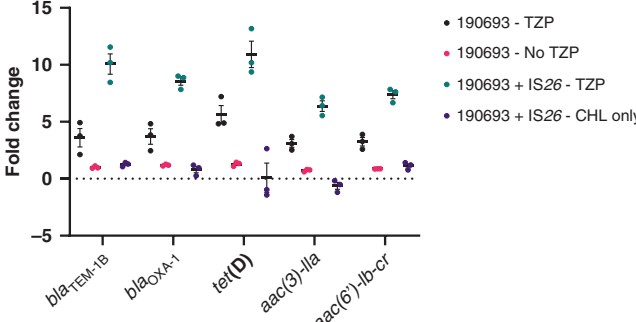

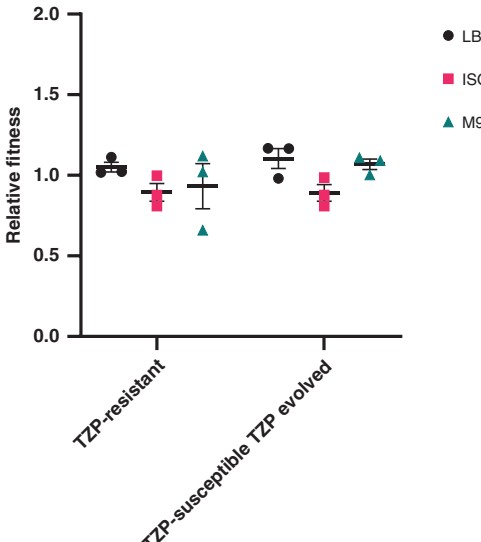

**Fig. 3 Copy number of antimicrobial resistance genes present on the translocatable unit increased following in vitro replication of the evolutionary event leading to piperacillin/tazobactam resistance.** Fold change in copy number of all the antimicrobial resistance genes found on the pseudo-compound transposon compared to the housekeeping gene *uidA* following growth of the piperacillin/tazobactam (TZP)-susceptible isolate in the absence of antibiotics (pink circles), TZP-susceptible isolate in the presence of 8/4 µg/ml TZP (black circles), TZP-susceptible isolate transformed with pHSG396 plasmid containing IS*26* in the presence of 8/4 µg/ml TZP and 35 µg/ml chloramphenicol (green circles) and TZP-susceptible isolate transformed with pHSG396 plasmid containing IS*26* in the presence of 35 µg/ml chloramphenicol only (purple circles). All error bars represent the standard error of the mean ($n = 3$). Source data are provided as a Source Data file.

and 1385 bp (right) products (Fig. 2d), consistent with the insertion of the TU adjacent to IS*26* and reforming PTn*6762* in pHSG396 (Supplementary Fig. 5B). Subsequent long-read sequencing of the entire pHSG396:IS*26* plasmid following the capture of the TU established the presence of at least three different plasmid structures; pHSG396:IS*26* without the captured TU (Supplementary Fig. 6A), pHSG396:IS*26* with the captured TU (Supplementary Figs. 6B and 7) and tandem pHSG396:IS*26* with TU (Supplementary Figs. 6C and 7B). These structures were confirmed by the presence of sequencing reads which spanned the respective junctions of pHSG396:IS*26* and the captured TU. In both plasmid structures containing the captured TU, at least one captured TU had the same structure as predicted by the hybrid assembly and was inserted adjacent to the copy of IS*26* cloned into pHSG396, reforming the pseudo-compound transposon PTn*6762*. The only variation was seen in the tandem pHSG396:IS*26* plus TU, where *bla*OXA-1 was mutated in one of the captured TUs (Supplementary Fig. 7B). This was due to a single nucleotide insertion resulting in a frame shift and a terminal stop to the protein.

**Replication of excision of the TU.** We sought to determine whether the presence of another mobile IS*26* and/or the use of TZP induced the excision of the TU by replicating the evolutionary event that led to the TZP-susceptible isolate becoming resistant to TZP in vitro. We found that there was a significant increase in copy number of all the resistance genes present on PTn*6762*, relative to the housekeeping gene *uidA*, following exposure to 2 × MIC of TZP (8/4 µg/ml) while there was no evidence of amplification in the same isolate grown in the absence of TZP (*P* value = <0.0001, Fig. 3, Supplementary Table 3). There was also an increase in copy number when the isolate containing the pHSG396:IS*26* plasmid was exposed to TZP but, again, no increase in copy number when TZP was absent (*P* value = <0.0001, Fig. 3, Supplementary Table 3), therefore TZP can either select for the maintenance of the excised TU or induce the excision event, leading to an increase in copy number of the genes present on PTn*6762*. In contrast, there was no significant difference in gene copy number between the TZP-susceptible isolate

**Fig. 4 Gene amplification did not incur a fitness cost in the piperacillin/tazobactam-resistant isolate compared to the piperacillin/tazobactam-susceptible isolate.** Relative fitness of the piperacillin/tazobactam (TZP)-resistant isolate compared to the TZP-susceptible isolate and the TZP-susceptible isolate grown in the presence of TZP compared to the TZP-susceptible isolate grown in the absence of TZP, assessed comparatively in LB broth (LB, black circles), iso-sensitest broth (ISO, pink squares) and M9 (green triangles). All error bars represent the standard error of the mean ($n = 3$). Source data are provided as a Source Data file.

with and without the pHSG396:IS*26* plasmid grown in the absence of TZP (*P* value = 0.1355, Fig. 3, Supplementary Table 3), underlining that the presence of an extra chromosomal IS*26* does not induce excision of the TU from PTn*6762*.

**Fitness effect of extensive amplification.** Hyperproduction of a protein could result in a fitness cost to the cell due to the increased metabolic activity. Yet, we found that the amplification of the TU and carriage of the large plasmid in the TZP-resistant isolate did not result in a significant change in fitness compared to the TZP-susceptible isolate in LB broth (LB, *P* value = 0.6523), iso-sensitest (ISO, *P* value = 0.3587) and M9 (*P* value = 0.5532, Fig. 4, Supplementary Table 3). We also assessed the relative fitness of the TZP-susceptible isolate following exposure to TZP (resulting in the increase in copy number of the resistance genes present on the TU (Fig. 3)) compared to the TZP-susceptible grown in the absence of TZP (which did not result in an increase in copy number (Fig. 3)). Again, we found no significant change in fitness in LB (*P* value = 0.1264), ISO (*P* value = 0.1126) and M9 (*P* value = 0.3007, Fig. 4, Supplementary Table 3).

## Discussion

Tazobactam is able to inhibit the activity of class A β-lactamases[32], and therefore the presence of *bla*TEM-1 within the genome of an *E. coli* isolate should not result in resistance to TZP. However, two studies have linked amplification of *bla*TEM, and therefore hyperproduction of the β-lactamase, with this phenotype[17,20], with one linking amplification and the presence of IS*26*[20]. Although, the exact mechanism of amplification/hyperproduction has remained elusive. Due to the emergence of a TZP-resistant but 3rd generation cephalosporins and carbapenems susceptible *E. coli* and *K. pneumoniae* phenotype[12–14], as well as the increasing reliance on TZP as an empirical treatment[6] and the recent interest in the use of TZP as a carbapenem sparing treatment for extended-spectrum β-lactamase infections[33,34], it is of growing importance to understand the

mechanism of amplification. In this study we had the opportunity to compare a pair of clonal isolates which have evolved within a patient to become TZP-resistant but remain cephalosporin/carbapenem susceptible, allowing us to build on recent studies and identify the mechanism of IS26-mediated amplification of $bla_{TEM-1B}$ which leads to TZP-resistance.

We found multiple antibiotic resistance genes, including $bla_{TEM-1B}$, co-located on a IS26-flanked pseudo-compound transposon PTn6762 on both the TZP-resistant and TZP-susceptible isolate. The resistance genes $tet$(D) and $bla_{TEM-1B}$ were both present on PTn6762 as a result of the insertion of several transposons into the same location, demonstrated by the presence of multiple insertion sequences and transposable elements on PTn6762. While tazobactam has intermediate inhibitory activity towards OXA-1[16], we found that the presence of this β-lactamase in the TZP-susceptible isolate did not confer resistance to TZP, amplification of $bla_{OXA-1}$ was involved not in resistance to TZP and TEM-1B is likely the most important determinant of TZP resistance. Although we were able to determine that efflux pumps did not have a role in TZP-resistance in this isolate, we did not investigate whether other mechanisms, such as change in permeability due to porins, contributed to TZP-resistance. Amplification of $bla_{TEM-1B}$ is achieved when PTn6762 is excised from the chromosome forming a TU (Fig. 5), evidenced by the hybrid assembly of the TZP-resistant isolate, increase in copy number of the antibiotic resistance genes present on PTn6762 and the capture of the TU in pHSG396:IS26. While it is a possibility that the presence of the circularised TU in the hybrid assembly is due to an artefact of the assembly, we confirmed the presence of the excised TU following capture in pHSG396:IS26. The TU then re-inserts into the chromosome adjacent to a copy of IS26 to create a tandem array of TUs, increasing the copy number of the antibiotic resistance genes present on the TU (Fig. 5). While precise excision and formation of a TU containing the antibiotic resistance gene $aphA1a$ from a Tn4352B transposon present in a plasmid has been demonstrated before as a mechanism of movement of antibiotic resistance gene[23,24,35], we have shown that this mechanism can directly result in gene amplification leading to antibiotic resistance. We found no

evidence from the whole-genome sequencing of the TZP-resistant isolate of insertion of the TU anywhere else in the chromosome, except for a gap in sequencing where PTn6762 was originally situated in the TZP-susceptible isolate adjacent to an IS26. Through PCR of the left and right junctions of the gap in sequencing, we confirmed that PTn6762 was still present at this location in the chromosome of the TZP-resistant isolate and therefore existed as PTn6762 with tandem repeats of the TU in the chromosome, which the hybrid assembly was unable to resolve. Rodriguez-Villodres et al.[36] replicated the amplification event leading to TZP-resistance in $E. coli$ isolates containing $bla_{TEM}$ through exposure to increasing concentrations of TZP[36]. They found an increased copy number of $bla_{TEM}$ in TZP-resistant isolates and regions of amplification, in which one of the amplified regions contained IS26[36]. Using the CHL-resistant pUC vector, pHSG396, containing a copy of IS26 from the TZP-susceptible chromosome, we were also able to replicate the evolutionary event and capture the excised TU adjacent to the IS26 copy in the plasmid, providing evidence of the insertion event, but we found no evidence of tandem repeats of the TU in the pHSG396:IS26 plasmid. Therefore, the TU preferentially re-inserts into the chromosome adjacent to the chromosomally located IS26 via a conservative Tnp26-dependent but RecA-independent mechanism, Tnp26 replicative transposition or RecA-dependent homologous recombination[24] producing tandem repeats of the TU. Again, while this mechanism has previously been demonstrated in terms of movement of antibiotic resistance genes[23,24], we have shown that this mechanism will cause gene amplification resulting in antibiotic resistance in a clinical isolate. This mechanism is of concern as IS26 has been associated with the transfer of $bla_{NDM-1}$, in a recent nosocomial outbreak in Germany[35], and other carbapenemases[37] and therefore represents a risk of clinical resistance to any carbapenemase inhibitor currently in development[38] which needs to be investigated further. The method of capture of the TU used in this study can be used to investigate whether the same mechanism will result in gene amplification and subsequent reduction in efficacy to other β-lactam/β-lactamase inhibitors, as well as to further confirm the role of TZP in the induction of excision of the TU, and therefore gene amplification. Abdelraouf et al.[15] identified five TZP-resistant but 3rd generation cephalosporin susceptible $K. pneumoniae$ which did not contain $bla_{TEM}$, but instead, the most common type of β-lactamase was SHV-1[15]. These isolates showed a different phenotypic resistance profile to the $E. coli$ isolates containing $bla_{TEM}$ assessed in the same study; exhibiting higher MICs against the 3rd generation cephalosporins tested and susceptibility to TZP was not restored with increasing concentrations of tazobactam, potentially indicating a different mechanism of resistance[15]. Further work should be undertaken with both $E. coli$ and $K. pneumoniae$ to fully understand both the resistance profile and the different mechanisms of resistance involved in this phenotype to aid diagnostics.

Hansen et al.[20] associated amplification of $bla_{TEM-1}$ in an $E. coli$ clinical isolate with a significant fitness cost[20]; Fitness of this isolate was compared to other unrelated clinical isolates of $E. coli$ which hyperproduce TEM-1 due to promoter mutations, rather than the same isolate with and without amplification. This approach can lead to over- or under-estimation of fitness cost, as genetic background of the isolate can have an impact on the overall fitness affect, as can the environment fitness is assessed in[39,40]. Adler et al.[22], however, identified a fitness cost associated with IS26-mediated amplification of an antibiotic resistance cassette from a plasmid in all lineages tested[22]. In this study, we were able to compare the fitness of the paired clinical isolates and in vitro evolved isolates with and without amplification of the TU and found that there was no significant difference in fitness cost, despite amplification of a >10 kb region with multiple functionally transcribed genes and, in terms of the TZP-resistant clinical

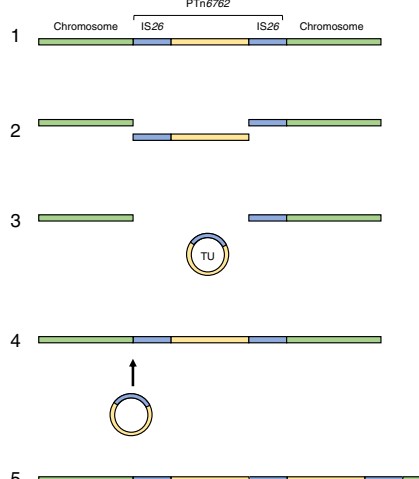

**Fig. 5 Proposed mechanism of hyperproduction of TEM-1B mediated by IS26.** Schematic of the proposed mechanism of amplification leading to hyperproduction of TEM-1B; (1) a pseudo compound transposon, PTn6762, is present on the chromosome flanked by two copies of IS26. (2) PTn6762 is excised from the chromosome, (3) which then forms a TU which (4) re-inserts into the chromosome adjacent to the chromosomally located IS26 (5) creating a tandem array of the TU and increasing the copy number of $bla_{TEM-1B}$.

isolate, has acquired a large plasmid. If this lack of fitness cost is translated into a physiological environment, it may result in the TZP-resistant phenotype persisting. While there was no observed fitness effect of amplification on the single isolate in this study, the effect of amplification on bacterial fitness needs to be extensively investigated as it may not be a global phenomenon

Resistance to the β-lactam/β-lactamase inhibitor TZP can be the result of gene amplification, and subsequent hyperproduction, of $bla_{TEM-1B}$. The mechanisms involved in this gene amplification are the IS26-associated excision and re-insertion into the chromosome, of a TU containing $bla_{TEM-1B}$ derived from a chromosomally located pseudo-compound transposon PTn6762, which is either selected by or in response to exposure to TZP. The TU is capable of re-inserting into the chromosome, creating a tandem array of TUs and increasing the copy number of $bla_{TEM-1B}$. In this clinical isolate and an in vitro evolved isolate, we found that there was no effect on fitness due to the amplification and subsequent carriage of high numbers of the TU. This mechanism of amplification, and the subsequent hyperproduction, of $bla_{TEM-1B}$ is an important consideration if treatment failure involving TZP occurs, as well as other β-lactam/β-lactamase inhibitor combinations, and when using genomic data to predict resistance/susceptibility to β-lactam/β-lactamase inhibitor combinations.

## Methods

**Ethics statement**. All *E. coli* isolates used in this study were collected at the RLUH (Liverpool, UK) as part of routine clinical diagnostics procedures. The bacterial isolates were identified in the hospital biobank database as having interesting resistance profiles by the Consultant Microbiologist. Isolates were retrieved from the Microbiology Laboratory by Thomas Edwards (Liverpool School of Tropical Medicine), who has an NHS research passport enabling work in the hospital laboratories. Antimicrobial susceptibility data and the treatment data were anonymised, unlinked to patient identifiers and data produced in this study was not used for the treatment or management of patients, therefore requirement for ethical approval and informed patient consent was not required. This was confirmed using the online NHS Research Ethics Committee review tool http://www.hra-decisiontools.org.uk/ethics/.

**Bacterial isolates, media and antibiotics**. Clinical isolates of *E. coli* isolated from blood cultures between 2010 and 2017 at the RLUH (Liverpool, UK) which were found to be carbapenem and cephalosporin susceptible but TZP resistant using the disk diffusion method of antimicrobial susceptibility testing (AST) were initially identified from isolate records. Isolate records were then searched for a corresponding carbapenem, cephalosporin and TZP susceptible isolates, isolated in the same or a previous infection episode from the same patient. Using these criteria, we identified five paired clinical isolates of *E. coli*. All isolates had been stored at the time of blood culture isolation in glycerol broth at −80 °C.

All isolates were grown on LB (Lennox) agar at 37 °C for 18 h followed by growth in LB, (Sigma, UK), ISO (Oxoid, UK) or M9 (50% (v/v) M9 minimal salts (2x) (Gibco, ThermoFisher Scientific, USA), 0.4% D-glucose, 4 mM magnesium sulphate (both Sigma, UK) and 0.05 mM calcium chloride (Millipore, USA)) at 37 °C for 18 h at 200 rpm.

Piperacillin, tazobactam (both Cayman Chemical, USA), gentamicin (GEN), and amoxicillin trihydrate:potassium clavulanate (4:1, AMC) was solubilised in molecular grade water (all Sigma, UK), while CHL and tetracycline (TET) (both Sigma, UK) were solubilised in ethanol (VWR, USA) and ciprofloxacin (CIP) was solubilised in 0.1 N hydrochloric acid solution (both Sigma, UK). All stock solutions of antibiotics were filter sterilised through a 0.22 μm polyethersulfone filter unit (Millipore, USA). In all assays, unless stated, tazobactam was used at a consistent concentration of 4 μg/ml and the piperacillin concentration was altered.

**Restriction enzyme digestion**. RFLP analysis of 1 μg of 16S rRNA PCR amplicon from the 10 putative clonal isolates were digested with AlwNI, PpuHI and MslI (all New England Biolabs (NEB), USA) and 1 μg of long fragment genomic DNA extracts of 153964, 152025, 190693 and 169757 were digested with SpeI and MslI (both NEB, USA) for 1 h at 37 °C. Both RFLP digest reactions were incubated for 5 min at 80 °C and immediately run on a 1% agarose gel. Enzyme digest of 500 ng of plasmid DNA extracted from 190693 and 169757 was performed with PpuMI and XhoI (both NEB, USA) and immediately run on a 1% agarose gel following incubation for 1 h at 37 °C. RFLP was performed on a single DNA extract from each bacterial isolate.

**Antimicrobial susceptibility testing**. Initial AST for cefpodoxime (CPD), cefoxitin (FOX), TZP, meropenem (MEM), CIP, cefotetan (CTT), amikacin (AMK),

ertapenem (ETP), AMC, CHL and ampicillin (AMP) was performed in RLUH clinical laboratory using the disk diffusion method, according to either British Society for Antimicrobial Chemotherapy (BSAC) or EUCAST guidelines for Antimicrobial Susceptibility Testing[41–43].

MIC for TZP, GEN, CIP, CHL, AMC and TET were performed using the broth microdilution method, in cation adjusted Mueller Hinton Broth (CA-MHB, Sigma, UK), following EUCAST Guidelines[44]. Efflux pump inhibition was performed using PAβN as a supplement in CA-MHB at a final concentration of 50 μM, inhibition of TEM-1B was determined with piperacillin with increasing concentrations of tazobactam (4, 8 and 16 μg/ml) and inhibition of OXA-1 was determined with TZP plus 100 mM sodium chloride (Sigma, UK). Each MIC was performed in triplicate using independent bacterial cultures, each with three technical replicates.

**Nitrocefin assay**. β-lactam hydrolysis was evaluated using a colorimetric nitrocefin assay. Cell lysates were obtained from triplicate cultures of 190693 and 169757 in LB, adjusted to an optical density at 600 nm ($OD_{600}$) of 0.1. Cultures (10 ml) were centrifuged at $14,000 \times g$ for 5 min, the supernatant discarded, and the pellet resuspended in 5 ml phosphate-buffered saline (PBS). The cultures were sonicated for three intervals of ten seconds, on ice, using a Soniprep 150 plus (MSE centrifuges, UK). The lysed cultures were centrifuged at $14,000 \times g$ for 5 min, and the supernatant taken as the culture lysate.

A total of 90 μl of this lysate was then added to 10 μl of 0.5 mg/ml nitrocefin solution (Sigma, UK) in a 96-well microplate, in triplicate. The absorbance of the plate was read at an optical density of 450 nm ($OD_{450}$) every 20 s for 25 min, using a SPECTROstar OMEGA spectrophotometer (BMG lab systems, Germany). This assay was performed in triplicate using independent bacterial cultures.

**Whole-genome sequencing and bioinformatics**. Whole-genome sequencing was performed from a single DNA extract for each isolate. Illumina MiSeq $2 \times 250$ bp short-read sequencing of long fragment DNA extractions from isolates 190693 and 169757, as well as adapter trimming of the sequencing reads, were provided by MicrobesNG (MicrobesNG, UK).

The same long fragment DNA extracts were processed using the SQK-LSK109 ligation and SQK-RBK103 barcoding kit and sequenced on an R9.4.1 flow cell with an ONT MinION. Sequencing reads were basecalled during the sequencing run using MinKNOW (v19.05.0), de-multiplexing and adapter trimming of the basecalled reads were performed using Porechop (v0.2.4) and finally sequencing reads were filtered for a quality score of 10 via Filtlong (v0.2.0).

Both Illumina short-read and ONT long-read sequences were assembled using Unicycler (v0.4.7[45]), with the quality of the assembly assessed using QUAST (v5.0.2[46]), annotated using Prokka (v1.14.0[47]) and visualised using Bandage (v0.8.1[48]).

Sequence type and serotype of both 190693 and 169757 were determined using Multi-Locus Sequence Typing (MLST, v2.0.4[49]) and SerotypeFinder (v2.0.1[50]), respectively. The relatedness of the two genomes were compared using MUMmer (v3.23[51]) and the ANI was calculated using OrthoANI (v0.93.1[52]). Presence of acquired antimicrobial resistance genes within the two genomes were assessed using ResFinder with minimum threshold of 90% and a minimum length of 60% (v3.2[53]) and segments of the two genomes were characterised using SnapGene® software (v3.3.4, from GSL Biotech; available at snapgene.com). Finally, plasmid replicons were identified using PlasmidFinder (v2.0.1[54]).

Long-read sequencing reads of 190693 and 169757 were mapped to a predicted structure of three TUs in tandem using BWA MEM[55] (https://github.com/lh3/bwa, v0.7.17-r1188). Aligned sequencing reads were converted to FASTQ using samtools (ref. [56] v1.10) and then used to a build a consensus sequence with Medaka (https://github.com/nanoporetech/medaka, v0.11.5) and compared to the predicted tandem TU structures in EasyFig (ref. [57] v2.2.2).

**Competent cell preparation**. The TZP-susceptible isolate was made competent according to Chung et al.[58].

**Quantitative PCR**. Changes in gene copy number of $bla_{TEM-1B}$, $bla_{OXA-1}$, $aac(3)$-$lla$, $aac(6')$-$lb$-$cr$, $tet(D)$ were calculated via qPCR, using the ΔΔCT method for relative quantitation of these genes against the single copy $uidA$ housekeeping gene.

Each qPCR reaction contained 6.25 μl QuantiTect® SYBR Green PCR buffer (Qiagen, UK), 0.4 μM forward and reverse primers (Supplementary Table 4), 1 ng of extracted DNA, and molecular grade water to a final volume of 12.5 μl. Reactions were processed using a Rotor-Gene Q (Qiagen, Germany), using the following protocol; an initial denaturation step of 95 °C for 5 min, followed by 40 cycles of; DNA denaturation at 95 °C for 10 s, primer annealing at 58 °C for 30 s, and primer extension at 72 °C for 10 s with fluorescence monitored in the FAM channel. High resolution melt analysis was carried out over a temperature range of 75 °C to 90 °C, increased in 0.1 °C increments, in order to confirm specific amplification. Fluorescence thresholds were set manually for calling Ct values, at 5% of the difference between baseline and maximum fluorescence.

The mean qPCR Ct value for the $uidA$ gene from each strain was taken using at least three replicates qPCR reactions, and the ΔΔCT method was utilised to determine to fold change using at least triplicate qPCR reactions for each AMR gene from a single DNA extract of each isolate.

**In vitro evolution of susceptible isolate**. The clinical isolate 190693 (TZP-susceptible isolate) and 190693 transformed with pHSG396:IS26 were subcultured into 10 ml LB and 10 ml LB plus 35 μg/ml CHL, respectively, and incubated at 37 °C for 18 h at 200 rpm. Following incubation 10 μl of 190693 was subcultured into 10 ml LB and 10 ml LB plus 8/4 μg/ml TZP and 10 μl of 190693 with pHSG396:IS26 was subcultured into 10 ml LB plus 35 μg/ml CHL and 10 ml LB plus 35 μg/ml CHL and 8/4 μg/ml TZP and incubated at 37 °C for 24 h at 200 rpm. Each evolution experiment was performed in triplicate using independent cultures. Genomic DNA from each of the four cultures from each replicate were extracted for qPCR following the protocol described in Supplementary Methods 1, except a single replicate of each of the three biological replicates were performed.

**TU capture**. The TZP-susceptible isolate 190693 transformed with pHSG396:IS26 was grown in the presence of TZP and CHL as previously stated. Following selection, the culture was serially diluted 1/10 in PBS down to $10^{-7}$ dilution and 50 μl of each dilution was plated out on to LB agar supplemented with 35 μg/ml CHL and 16/4 μg/ml TZP. Five single colonies were selected and subcultured into 10 ml LB plus 35 μg/ml CHL and 16/4 μg/ml TZP for 18 h at 37 °C and 200 rpm and the plasmid extracted following the protocol in Supplementary Methods 1. The purified plasmids were transformed into NEB® 5-alpha competent E. coli (NEB, US) following the protocol in Supplementary Methods 1 and plated out on to LB agar supplemented with 35 μg/ml CHL and 16/4 μg/ml TZP and incubated at 37 °C for 18 h. A single colony from each transformation was subcultured into 10 ml LB supplemented with 35 μg/ml CHL and 16/4 μg/ml TZP and incubated at 37 °C, 200 rpm for 18 h and the plasmid extracted following the protocol in Supplementary Methods 1. The initial pHSG396:IS26 plasmid extract and pHSG396: IS26 plasmid selected in TZP and extracted from NEB® 5-alpha E. coli were digested with XhoI (NEB, US) and EcoRI for 1 h at 37 °C, followed by a 20 min incubation at 65 °C and run on a 1% agarose gel. A single DNA extract of pHSG396:IS26 selected in TZP was sequenced on a MinION, adapters removed, and sequencing reads filtered as stated above, except using a quality score of 50. Long-read sequencing reads of the captured TU in pHSG396:IS26 were mapped to three predicted plasmid structures, pHSG396:IS26, pHSG396:IS26 plus TU and tandem pHSG396:IS26 plus TU and used to build a consensus sequence using the method described above.

**Comparative fitness**. The relative fitness of 169757 (TZP-resistant) and 190693 (TZP-susceptible) grown in the presence of 8/4 μg/ml TZP, compared to 190693 and 190693 grown in the absence of TZP, respectively, were assessed comparatively in LB, ISO and M9. Each culture was diluted to an $OD_{600}$ of 0.1 in the respective media, then further diluted 1/1000 in the same media and 150 μl of each diluted culture added to a flat bottom, 96-well microtitre plate in duplicate as well as 150 μl of the media as a negative control. The 96-well plate was incubated at 37 °C and the $OD_{600}$ of each well was measured with 100 flashes every 10 min over 24 h, with orbital shaking at 200 rpm between readings, using a Clariostar Plus microplate reader (BMG Labtech, Germany). The relative fitness compared to either 190693 or 190693 grown in the absence of TZP between absorbance values 0.02 and 0.08 and a minimum R value of 0.9905 was estimated using BAT version 2.1[59]. Comparative fitness was performed in triplicate with independent bacterial cultures, with two technical replicates for each replicate.

**Statistical analysis**. Statistical analysis of comparison for the qPCR assay of the antibiotic resistance genes was performed using the two-way ANOVA with Uncorrected Fisher LSD test. Statistical analysis of the nitrocefin assay was performed using the two-way ANOVA test. Statistical analysis of relative fitness of 169757 and 190693 grown in the presence of TZP was performed using ordinary one-way ANOVA with Uncorrected Fisher LSD test. All statistical tests were performed using GraphPad Prism version 8.2.1. Means and standard error of the mean of all data presented in this study are available in Supplementary Table 3.

**Reporting summary**. Further information on research design is available in the Nature Research Reporting Summary linked to this article.

## Data availability
All sequencing reads and assemblies were deposited to GenBank under the BioProject number PRJNA607545. Consensus sequences of pHSG396:IS26, pHSG396:IS26 plus translocatable unit (TU) and tandem pHSG396:IS26 plus TU are available in Supplementary Fig. 6A–C. Source data are provided with this paper.

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

## Acknowledgements

This work was supported by the Liverpool School of Tropical Medicine Director's Catalyst Fund awarded to A.T.M.H. and T.E. A.P.R. would like to acknowledge funding from the AMR Cross-Council Initiative through a grant from the Medical Research Council, a Council of UK Research and Innovation (Grant number; MR/S004793/1), and funding from the National Institute for Health Research. (Grant Number; NIHR200632). This report is independent research funded by the Department of Health and Social Care. The views expressed in this publication are those of the authors and not necessarily those of the NHS or the Department of Health and Social Care.

## Author contributions

A.T.M.H. and T.E. conceptualised the study. J.M., P.R., C.M.P., C.C., J.v.A., and A.H. collated isolate metadata, clinical antimicrobial susceptibility data and patient treatment data. A.T.M.H., A.J.F., E.R.A., A.P.R. and T.E. contributed to the experimental design and data analysis. A.T.M.H., I.B., A.J.F. and T.E. contributed to carrying out the experiments. A.T.M.H. and T.E. wrote the first draft of the manuscript, which was then edited and approved all authors by all authors.

## Competing interests

The authors declare no competing interests.
