## [Peer Review File · Nature Communications]

REVIEWER COMMENTS

Reviewer #1 (Remarks to the Author):

The manuscript of Hubbard et al., investigates the evolution of TZP resistance in two clinical *E. coli* isolates obtained 3 months apart from the same patient. They establish the relatedness of the two isolates by RFLP and WGS. WGS by both short-read and long-read technology associated the development of TZP resistance to an increase in coverage of an approximately 10 kb DNA segment containing two beta-lactamases, blaOXA-1 and blaTEM-1. The authors are able to show an concomitant increase in corresponding mRNA levels and nitrocefin conversion levels. Hybrid assembly using short and long reads suggests the existence of the amplified gene segment as an excised circular structure compatible with a TU (transposable unit) element as described by Harmer et al., 2014. Furthermore, they are able to capture a TU element in their replication of resistance development using a IS26-containing element as target in a experiemntal setup very reminiscent of the setup of Harmer et al., MBio 2014.

The paper is overall interesting, but this reviewer also has some concerns:

Both in the title and throughout the manuscript, the authors emphasizes the importance of amplification of blaTEM-1. Compared to the isolates reported by Hansen et al., JAC 2019 amplification of blaTEM-1 is relatively modest. However, the isolates reported here also contain a blaOXA-1, an enzyme previously reported to be of importance to TZP resistance (e.g. Livermore et al., JAC 2019). The relative importance of the two enzymes in TZP resistance may possibly be determined by choice of and increasing concentration of beta-lactamase inhibitor.

The in vitro development of TZP resistance by gene duplication has previously been shown by Rodriquez-Villodres et al, JAC 2020 - a study the authors may wish to discuss. In the present study, the novel claim is that the amplification is maintained by a TU, a somewhat surprising finding given that the TU does not contain a replication origen. The authors uses primers RM2_F2 and RM_R1 (Fig 2B and 2C) in an effort to verify the existence of the circular TU structure. Would the same primers yield the same PCR products if the amplification was organized in a tandem arrays as found by others? I encourage the authors to reanalyze their ONT reads in order to verify the presence or absence of repeat structures. This analysis is of course pendent on a sufficient number of reads longer than 10 kb.

The authors report the acquisition of a plasmid in the TZP resistant clinical isolate. This contains an IS26 element and may therefore be able to capture a TU. The authors speculate that this may not have occurred because of an orienntaion effect of the IS26 element. Such an effect may be experimentally pursued using a derivative of pHSG396:IS26. I missed more detailed information on pHSG396 in the manuscript.

Minor points:

Line 85: As I recall, the paper of Schechter et al. reports a tandem amplification of blaTEM-1 on a plasmid, please verify.

Line 302: Consulting Figure 2A I am in doubt if the circular molecule only contains a single IS26 element.

Line 351: I suggest to sequence the entire plasmid to verify the structure and length of the captured TU.

Throughout the manuscripts the authors reports P-values only. I suggest to include also averages and SE; not all statistically significant findings may translate into biological importance.

Reviewer #2 (Remarks to the Author):

Overall a terrific bit of work, wonderful to be able to go back in clinical micro stocks to find clonal pairs from same patient with desired phenotypic profile to TZP.

1. I didn't note any in vitro expression studies to assess functional status of mutations as noted? At the very least, additional MIC studies should be conducted with varying [c] of tazobactam to ensure function/inhibition.

2. Line 148: notes CLSI methods, line 269 notes EUCAST?

3. Agree with your assessment of fitness. Compare and contrast to recent publication by the original reporters of this novel phenotype using mouse sepsis model that also suggests this phenotype produces the expectedly high in vivo mortality when untreated [Piperacillin-Tazobactam-Resistant/Third-Generation Cephalosporin-Susceptible *Escherichia coli* and *Klebsiella pneumoniae* Isolates: Resistance Mechanisms and In vitro-In vivo Discordance. Abdelraouf K, Chavda KD, Satlin MJ, Jenkins SG, Kreiswirth BN, Nicolau DP. *Int J Antimicrob Agents*. 2020 Mar;55(3):105885. doi: 10.1016/j.ijantimicag.2020.105885. Epub 2020 Jan 8.

4. TZP phenotype of interest has been observed in both *E. coli* and *Klebsiella*. Recent IJAA paper noted above is consistent with TEM hyperexpression for *E. coli*; however, a different mechanism appears to be responsible in *Klebsiella*. Would be reasonable in discussion to highlight potential difference among species, additional work should be done in *Klebsiella* to fully delineate the drive of this phenotypic profile.

Reviewer #1 (Remarks to the Author):

The manuscript of Hubbard et al., investigates the evolution of TZP resistance in two clinical *E. coli* isolates obtained 3 months apart from the same patient. They establish the relatedness of the two isolates by RFLP and WGS. WGS by both short-read and long-read technology associated the development of TZP resistance to an increase in coverage of an approximately 10 kb DNA segment containing two beta-lactamases, blaOXA-1 and blaTEM-1. The authors are able to show an concomitant increase in corresponding mRNA levels and nitrocefin conversion levels. Hybrid assembly using short and long reads suggests the existence of the amplified gene segment as an excised circular structure compatible with a TU (transposable unit) element as described by Harmer et al., 2014. Furthermore, they are able to capture a TU element in their replication of resistance development using a IS26-containing element as target in a experimental setup very reminiscent of the setup of Harmer et al., MBio 2014.

The paper is overall interesting, but this reviewer also has some concerns:

Both in the title and throughout the manuscript, the authors emphasizes the importance of amplification of blaTEM-1. Compared to the isolates reported by Hansen et al., JAC 2019 amplification of blaTEM-1 is relatively modest. However, the isolates reported here also contain a blaOXA-1, an enzyme previously reported to be of importance to TZP resistance (e.g. Livermore et al., JAC 2019). The relative importance of the two enzymes in TZP resistance may possibly be determined by choice of and increasing concentration of beta-lactamase inhibitor.

The reviewer has made an excellent observation with this comment. We have assessed the MICs of the TZP-resistant isolate towards TZP with increasing concentrations of tazobactam (8 and 16 µg/ml) and in the presence of 100 mM sodium chloride, which has been shown to inhibit bla_{OXA-1}. We found the MIC to TZP decreased both in the presence of higher concentrations of tazobactam and but not in the presence of sodium chloride, suggesting that amplification of blaTEM-1B only is involved in TZP-resistance. We have included this new data in the manuscript and updated the text, all changes have been highlighted in yellow.

The in vitro development of TZP resistance by gene duplication has previously been shown by Rodriquez-Villodres et al, JAC 2020 - a study the authors may wish to discuss. In the present study, the novel claim is that the amplification is maintained by a TU, a somewhat surprising finding given that the TU does not contain a replication origin. The authors uses primers RM2_F2 and RM_R1 (Fig 2B and 2C) in an effort to verify the existence of the circular TU structure. Would the same primers yield the same PCR products if the amplification was organized in a tandem arrays as found by others? I encourage the authors to reanalyze their ONT reads in order to verify the presence or absence of repeat structures. This analysis is of course pendent on a sufficient number of reads longer than 10 kb.

We thank the reviewer for this comment and the suggestion of the recently published article, and we have included the article in the discussion. We are not claiming the TU amplifies by itself as it lacks an origin of replication, as the reviewer correctly pointed out. It is true that the two primers would yield the same product if the TUs were in a tandem array in the chromosome. Therefore, we mapped the long-read sequences to a predicted tandem (three) TU structure to determine if they were present in a tandem array. We found that there were no tandem TU structures in the TZP-susceptible isolate, while there was in the TZP-resistant isolate. This explains the mechanism of the amplification, whereby the TU exists as an extra chromosomal circular molecule which then re-inserts into the chromosome to produce a tandem array, thereby increasing the copy number of the β-lactamase.

We have updated the manuscript to reflect this (highlighted in yellow) and included an extra figure in the supplementary material.

The authors report the acquisition of a plasmid in the TZP resistant clinical isolate. This contains an IS26 element and may therefore be able to capture a TU. The authors speculate that this may not have occurred because of an orientation effect of the IS26 element. Such an effect may be experimentally pursued using a derivative of pHSG396:IS26. I missed more detailed information on pHSG396 in the manuscript.

Following further examination of the relevant literature we have concluded that the orientation of IS26 is unlikely to be a factor in the insertion of the TU, as the TU does not have an origin of replication. Therefore, the TU will be able to re-orientate itself to align with the two IS26 to facilitate insertion. Whilst it is possible that the insertion of the TU occurred in the plasmid, but due to the transient and rare nature of insertion it was missed during whole genome sequencing. Unlike the tandem array of TUs in the chromosome we were unable to find any sequencing reads that spanned the junctions of the TU and plasmid. Therefore, we have removed this paragraph from the manuscript. We have also included that pHSG396 is a chloramphenicol resistant, high copy number, pUC plasmid. We have updated the manuscript to reflect this, highlighted in yellow.

Minor points:

Line 85: As I recall, the paper of Schechter et al. reports a tandem amplification of bla_{TEM-1} on a plasmid, please verify.

We thank the reviewer for pointing out this error in the manuscript. We have updated the manuscript to clarify that there has been previously reported increase in copy number of bla_{TEM-1} located in either a chromosome or a plasmid, which has been highlighted in yellow in the manuscript.

Line 302: Consulting Figure 2A I am in doubt if the circular molecule only contains a single IS26 element.

Line 351: I suggest to sequence the entire plasmid to verify the structure and length of the captured TU.

We are treating these two comments as a single comment. Following long-read sequencing (by ONT) of the capture plasmid we had difficulty assembling the reads with long-read assemblers, so instead mapped to the sequencing reads to three predicted plasmid structures; pHSG396:IS26 with and without the TU and a tandem pHSG396:IS26 plus TU. All three predicted plasmid structures were present and in all plasmid structures containing the captured TU, the structure of the TU remained consistent and had inserted adjacent to the cloned copy of IS26 and completed the composite transposon in the plasmid. This therefore confirms that the TU only contains a single copy of IS26. We have updated the manuscript to reflect this and included a figure in the supplementary material to show the plasmid structures containing the TU.

Throughout the manuscripts the authors reports P-values only. I suggest to include also averages and SE; not all statistically significant findings may translate into biological importance.

We agree with the reviewer on reporting the means and standard error of the means of all data presented in this manuscript. Therefore, for complete transparency we have included all means and standard error of the mean for all data presented in this manuscript in Table S2.

Reviewer #2 (Remarks to the Author):

Overall a terrific bit of work, wonderful to be able to go back in clinical micro stocks to find clonal pairs from same patient with desired phenotypic profile to TZP.

1. I didn't note any in vitro expression studies to assess functional status of mutations as noted? At the very least, additional MIC studies should be conducted with varying [c] of tazobactam to ensure function/inhibition.

We have assessed the MICs of the TZP-resistant isolate towards TZP with increasing concentrations of tazobactam (8 and 16 µg/ml). We found the MIC to TZP decreased in the presence of higher concentrations of tazobactam confirming the functionality of bla_{TEM-1B}. We have included this new data in the manuscript and updated the text to reflect this, all changes have been highlighted in yellow.

2. Line 148: notes CLSI methods, line 269 notes EUCAST?

We thank the reviewer for this observation and the text has been updated. BSAC and EUCAST methods were in fact used in the clinical laboratory for disk diffusion testing, while EUCAST guidelines were used at LSTM for broth microdilution method. The manuscript has been updated to reflect this.

3. Agree with your assessment of fitness. Compare and contrast to recent publication by the original reporters of this novel phenotype using mouse sepsis model that also suggests this phenotype produces the expectedly high in vivo mortality when untreated [Piperacillin-Tazobactam-Resistant/Third-Generation Cephalosporin-Susceptible Escherichia coli and Klebsiella pneumoniae Isolates: Resistance Mechanisms and In vitro-In vivo Discordance. Abdelraouf K, Chavda KD, Satlin MJ, Jenkins SG, Kreiswirth BN, Nicolau DP. Int J Antimicrob Agents. 2020 Mar;55(3):105885. doi: 10.1016/j.ijantimicag.2020.105885. Epub 2020 Jan 8.

We thank the reviewer for the recommendation of the article. We have included in the discussion the author's assessment of fitness through survivability in the murine sepsis model and that this agrees with our assessment of fitness in our study. However, a more in-depth study is needed to understand the fitness effect of this phenotype. Changes to the manuscript have been highlighted in yellow.

4. TZP phenotype of interest has been observed in both E. coli and Klebsiella. Recent IJAA paper noted above is consistent with TEM hyperexpression for E. coli; however, a different mechanism appears to be responsible in Klebsiella. Would be reasonable in discussion to highlight potential difference among species, additional work should be done in Klebsiella to fully delineate the drive of this phenotypic profile.

We thank the reviewer for this suggestion and recommendation of this article. We completely agree with the reviewer's assessment of the potential different mechanisms of resistance between species and that further work to understand the different resistance profiles and mechanisms is needed. We have updated the discussion to reflect this, highlighted in yellow.

REVIEWERS' COMMENTS:

Reviewer #1 (Remarks to the Author):

The manuscript of Hubbard et al. describes the development of TZP resistance in vivo by comparing a pair of E.coli blood isolates obtained three months apart from a single patient. They establish the clonality of the two isolates and ascribe the developed resistance to amplification of a composite transposon Tn6762 containing blaTEM-1b and blaOXA-1 and other resistance genes. To determine the relative contribution of blaTEM-1 and blaOXA-1 they use inhibitors, sodium chloride and tazobactam.

Illumina and ONT sequencing and subsequent hybrid assembly shows Tn6762 to be located in the chromosome of the parental, TZP-susceptible isolate, but hybrid assembly in the resistant isolate fails to resolve the part of the chromosome where Tn6762 is located. Instead hybrid assembly results in three structures: a chromosome, a circular 10899 bp circular structure containing an IS26 element and a 530 bp IS26-derived structure. Additionally the isolate has acquired a 106637bp plasmid. Analysis of long reads from ONT verifies that Tn6762 is present as tandem repeats within the unresolved region of the chromosome.

The authors claim that the 10899 bp circular structure obtained by hybrid assembly present in a high copy number represents a translocable unit (TU). In support of the existence of the TU the authors are able to transform cells with a IS26-containing plasmid and capture the structure within the transformed plasmid. This is compatible with the in vitro experiments of Harmer et al. (ref 24). I will argue, however, that the 10899bp structure obtained by hybrid assembly could be an artefact of hybrid assembly when trying to resolve the chromosomal Tn6762 tandem array present in the by hybrid assembly unresolved region. It is a possibility that size distribution of long reads mapping to Tn6762 could provide evidence that sequencing actually captured a TU. In the lack of stronger evidence, I will suggest to discuss the possibility of the structure being an artefact of hybrid assembly of a tandem array and modify their manuscript accordingly.

Other comments:

Line 302-305: I appreciate the inclusion of inhibitors to discern between the contribution of OXA and TEM enzymes. I will suggest that the results are interpreted as amplification of blaTEM likely is the most important determinant of TZP resistance.

Line 352-354: In conjunction with the following lines a more likely conclusion would be that tandem amplifications were detected by the PCR.

Line 354-362: I believe this supports the tandem array in the chromosome. In line 442-444 you state that PCR confirmed the location of tandem arrays within the chromosome. I suggest these results is presented here. Did you go back to the ONT reads to confirm the "gap-bridging" PCR in the long reads?

line 364-372: Should include a reference to Fig 3. It is interesting to observe that amplification in the presence of TZP is more effective in the presence of an IS26-containing plasmid.

Line 389-391 and Fig S6B: I understand the tandem pHSG396:IS26 structure as a novel intra-plasmid tandem repeat. Is this correct? If so, please elaborate.

Line 432: This conclusion may be overreliant on hybrid assembly (cf. above).

Line 452: Is this compatible with line 389-91 and Fig S6B? Is it not just another tandem repeat observed in the plasmid?

Line 474: I believe Hansen et al. observed that passage in the absence of TZP led to a reduction of the tandem array and a gain of measured fitness.

Line 486-491: this argument uses virulence as a measure fitness. This may not be justified.

I appreciate the effort of the authors to have investigated the contribution of efflux pumps on TZP resistance and ruled these out. Could the authors extend their analyses to also include porins. It is remarkable that an 8-fold amplification has a profound effect; this could suggest that permeability also play a role in this set of isolates.

Reviewer #1 (Remarks to the Author):

The manuscript of Hubbard et al. describes the development of TZP resistance in vivo by comparing a pair of E.coli blood isolates obtained three months apart from a single patient. They establish the clonality of the two isolates and ascribe the developed resistance to amplification of a composite transposon Tn6762 containing blaTEM-1b and blaOXA-1 and other resistance genes. To determine the relative contribution of blaTEM-1 and blaOXA-1 they use inhibitors, sodium chloride and tazobactam.

Illumina and ONT sequencing and subsequent hybrid assembly shows Tn6762 to be located in the chromosome of the parental, TZP-susceptible isolate, but hybrid assembly in the resistant isolate fails to resolve the part of the chromosome where Tn6762 is located. Instead hybrid assembly results in three structures: a chromosome, a circular 10899 bp circular structure containing an IS26 element and a 530 bp IS26-derived structure. Additionally the isolate has acquired a 106637bp plasmid. Analysis of long reads from ONT verifies that Tn6762 is present as tandem repeats within the unresolved region of the chromosome.

The authors claim that the 10899 bp circular structure obtained by hybrid assembly present in a high copy number represents a translocable unit (TU). In support of the existence of the TU the authors are able to transform cells with a IS26-containing plasmid and capture the structure within the transformed plasmid. This is compatible with the in vitro experiments of Harmer et al. (ref 24). I will argue, however, that the 10899bp structure obtained by hybrid assembly could be an artefact of hybrid assembly when trying to resolve the chromosomal Tn6762 tandem array present in the by hybrid assembly unresolved region. It is a possibility that size distribution of long reads mapping to Tn6762 could provide evidence that sequencing actually captured a TU. In the lack of stronger evidence, I will suggest to discuss the possibility of the structure being an artefact of hybrid assembly of a tandem array and modify their manuscript accordingly.

We have acknowledged in the manuscript that the circularised TU may be due to an assembly artefact, but that the TU is present as we captured it in pHSG396:IS26, as shown by Oxford Nanopore sequencing of the plasmid::TU.

Other comments:

Line 302-305: I appreciate the inclusion of inhibitors to discern between the contribution of OXA and TEM enzymes. I will suggest that the results are interpreted as amplification of blaTEM likely is the most important determinant of TZP resistance.

We have altered this sentence to state that amplification of bla_{TEM-1B} is likely the most important determinant of TZP resistance as suggested.

Line 352-354: In conjunction with the following lines a more likely conclusion would be that tandem amplifications were detected by the PCR.

We have altered this sentence to conclude that the PCR detection of the TU showed that the TU was present in a tandem array in the chromosome, as suggested.

Line 354-362: I believe this supports the tandem array in the chromosome. In line 442-444 you state that PCR confirmed the location of tandem arrays within the chromosome. I suggest these results is presented here. Did you go back to the ONT reads to confirm the "gap-bridging" PCR in the long reads?

We have amended the manuscript to conclude the data presented supports that the TU exists in a tandem array in the chromosome of the TZP-resistant, as suggested. We did find evidence of sequencing reads spanning the junctions of the Tn6762/TU and chromosome, however they were not of sufficient length to encompass multiple TU sequences.

line 364-372: Should include a reference to Fig 3. It is interesting to observe that amplification in the presence of TZP is more effective in the presence of an IS26-containing plasmid.

We have included reference to Fig. 3 in this section as requested.

Line 389-391 and Fig S6B: I understand the tandem pHSG396:IS26 structure as a novel intra-plasmid tandem repeat. Is this correct? If so, please elaborate.

We are unsure of the exact structure the reviewer is referring to, so we have provided a diagram below of the structure we found;

Therefore, we did not find any intra-plasmid tandem repeat of translocatable unit in the plasmid structures following capture of the TU.

Line 432: This conclusion may be overreliant on hybrid assembly (cf. above).

We found the excision and movement of the TU is evident in the capture of the entire TU in pHSG396:IS26 and the presence of the TU in tandem repeats in the chromosome of the TZP-resistant isolate. Whilst potentially the observation that the TU forms a circular molecule could be an artefact of hybrid assembly, the existence of the TU and its physical capture in a plasmid, and its amplification has been proven in this manuscript.

Line 452: Is this compatible with line 389-91 and Fig S6B? Is it not just another tandem repeat observed in the plasmid?

Please see the diagram and response to the comment in regard to the tandem pHSG396:IS26 above. We did not detect any tandem repeats of the translocatable unit in pHSG396 following capture of the translocatable unit.

Line 474: I believe Hansen et al. observed that passage in the absence of TZP led to a reduction of the tandem array and a gain of measured fitness.

Hansen et al did observe an increase in fitness following a decrease in copy number of two evolved isolates (gene copy number 70 and 36) compared to the ancestral, unevolved isolate (gene copy number 182.6) following 150 generations of growth in the absence of antibiotics. However, there were no differences in fitness between the two evolved isolates despite an approximately 50% difference in gene copy number, suggesting the difference in fitness of the ancestor and evolved isolates is not entirely proportional to plasmid copy number and could be due to compensatory mutations developed during adaption to the growth media.

Line 486-491: this argument uses virulence as a measure fitness. This may not be justified.

The reviewer is correct and we have removed this from the manuscript.

I appreciate the effort of the authors to have investigated the contribution of efflux pumps on TZP resistance and ruled these out. Could the authors extend their analyses to also include porins. It is remarkable that an 8-fold amplification has a profound effect; this could suggest that permeability also play a role in this set of isolates.

We have included in the discussion that while we were able to rule out the role of efflux pumps in TZP-resistance in these isolates, we did not investigate the contribution of other mechanisms such as an increase in permeability due to porins.